# Robust Gaussian Graphical Modeling with the Trimmed Graphical Lasso

**Eunho Yang**
IBM T.J. Watson Research Center
eunhyang@us.ibm.com

**Aurélie C. Lozano**
IBM T.J. Watson Research Center
aclozano@us.ibm.com

## Abstract

Gaussian Graphical Models (GGMs) are popular tools for studying network structures. However, many modern applications such as gene network discovery and social interactions analysis often involve high-dimensional noisy data with outliers or heavier tails than the Gaussian distribution. In this paper, we propose the Trimmed Graphical Lasso for robust estimation of sparse GGMs. Our method guards against outliers by an implicit trimming mechanism akin to the popular Least Trimmed Squares method used for linear regression. We provide a rigorous statistical analysis of our estimator in the high-dimensional setting. In contrast, existing approaches for robust sparse GGMs estimation lack statistical guarantees. Our theoretical results are complemented by experiments on simulated and real gene expression data which further demonstrate the value of our approach.

## 1 Introduction

Gaussian graphical models (GGMs) form a powerful class of statistical models for representing distributions over a set of variables [1]. These models employ undirected graphs to encode conditional independence assumptions among the variables, which is particularly convenient for exploring network structures. GGMs are widely used in variety of domains, including computational biology [2], natural language processing [3], image processing [4, 5, 6], statistical physics [7], and spatial statistics [8]. In many modern applications, the number of variables $p$ can exceed the number of observations $n$. For instance, the number of genes in microarray data is typically larger than the sample size. In such high-dimensional settings, sparsity constraints are particularly pertinent for estimating GGMs, as they encourage only a few parameters to be non-zero and induce graphs with few edges. The most widely used estimator among others (see e.g. [9]) minimizes the Gaussian negative log-likelihood regularized by the $\ell_1$ norm of the entries (or the off-diagonal entries) of the precision matrix (see [10, 11, 12]). This estimator enjoys strong statistical guarantees (see e.g. [13]). The corresponding optimization problem is a log-determinant program that can be solved with interior point methods [14] or by co-ordinate descent algorithms [11, 12]. Alternatively neighborhood selection [15, 16] can be employed to estimate conditional independence relationships separately for each node in the graph, via Lasso linear regression, [17]. Under certain assumptions, the sparse GGM structure can still be recovered even under high-dimensional settings.

The aforementioned approaches rest on a fundamental assumption: the multivariate normality of the observations. However, outliers and corruption are frequently encountered in high-dimensional data (see e.g. [18] for gene expression data). Contamination of a few observations can drastically affect the quality of model estimation. It is therefore imperative to devise procedures that can cope with observations deviating from the model assumption. Despite this fact, little attention has been paid to robust estimation of high-dimensional graphical models. Relevant work includes [19], which leverages multivariate $t$-distributions for robustified inference and the EM algorithm. They also propose an alternative $t$-model which adds flexibility to the classical $t$ but requires the use of Monte Carlo EM or variational approximation as the likelihood function is not available explicitly. Another per-

tinent work is that of [20] which introduces a robustified likelihood function. A two-stage procedure is proposed for model estimation, where the graphical structure is first obtained via coordinate gradient descent and the concentration matrix coefficients are subsequently re-estimated using iterative proportional fitting so as to guarantee positive definiteness of the final estimate.

In this paper, we propose the *Trimmed Graphical Lasso* method for robust Gaussian graphical modeling in the sparse high-dimensional setting. Our approach is inspired by the classical Least Trimmed Squares method used for robust linear regression [21], in the sense that it disregards the observations that are judged less reliable. More specifically the Trimmed Graphical Lasso seeks to minimize a weighted version of the negative log-likelihood regularized by the $\ell_1$ penalty on the concentration matrix for the GGM and under some simple constraints on the weights. These weights implicitly induce the trimming of certain observations. Our key contributions can be summarized as follows.

- We introduce the Trimmed Graphical Lasso formulation, along with two strategies for solving the objective. One involves solving a series of graphical lasso problems; the other is more efficient and leverages composite gradient descent in conjunction with partial optimization.

- As our key theoretical contribution, we provide statistical guarantees on the consistency of our estimator. To the best of our knowledge, this is in stark contrast with prior work on robust sparse GGM estimation (e.g. [19, 20]) which do not provide any statistical analysis.

- Experimental results under various data corruption scenarios further demonstrate the value of our approach.

## 2   Problem Setup and Robust Gaussian Graphical Models

**Notation.**   For matrices $U \in \mathbb{R}^{p \times p}$ and $V \in \mathbb{R}^{p \times p}$, $\langle\!\langle U, V \rangle\!\rangle$ denotes the trace inner product $\mathrm{tr}(A\,B^T)$. For a matrix $U \in \mathbb{R}^{p \times p}$ and parameter $a \in [1, \infty]$, $\|U\|_a$ denotes the element-wise $\ell_a$ norm, and $\|U\|_{a,\mathrm{off}}$ does the element-wise $\ell_a$ norm only for off-diagonal entries. For example, $\|U\|_{1,\mathrm{off}} := \sum_{i \neq j} |U_{ij}|$. Finally, we use $\|U\|_\mathrm{F}$ and $\|U\|_2$ to denote the Frobenius and spectral norms, respectively.

**Setup.**   Let $X = (X_1, X_2, \ldots, X_p)$ be a zero-mean Gaussian random field parameterized by $p \times p$ concentration matrix $\Theta^*$:

$$\mathbb{P}(X; \Theta^*) = \exp\left( -\frac{1}{2} \langle\!\langle \Theta^*, X X^\top \rangle\!\rangle - A(\Theta^*) \right) \tag{1}$$

where $A(\Theta^*)$ is the log-partition function of Gaussian random field. Here, the probability density function in (1) is associated with $p$-variate Gaussian distribution, $N(0, \Sigma^*)$ where $\Sigma^* = (\Theta^*)^{-1}$.

Given $n$ i.i.d. samples $\{X^{(1)}, \ldots, X^{(n)}\}$ from high-dimensional Gaussian random field (1), the standard way to estimate the inverse covariance matrix is to solve the $\ell_1$ regularized maximum likelihood estimator (MLE) that can be written as the following regularized *log-determinant* program:

$$\underset{\Theta \in \Omega}{\text{minimize}} \ \left\langle\!\!\left\langle \Theta, \frac{1}{n} \sum_{i=1}^n X^{(i)}(X^{(i)})^\top \right\rangle\!\!\right\rangle - \log \det(\Theta) + \lambda \|\Theta\|_{1,\mathrm{off}} \tag{2}$$

where $\Omega$ is the space of the symmetric positive definite matrices, and $\lambda$ is a regularization parameter that encourages a *sparse* graph model structure.

In this paper, we consider the case where the number of random variables $p$ may be substantially larger than the number of sample size $n$, however, the concentration parameter of the underlying distribution is sparse:

**(C-1)** The number of non-zero off-diagonal entries of $\Theta^*$ is at most $k$, that is $|\{\Theta^*_{ij} \ : \ \Theta^*_{ij} \neq 0 \text{ for } i \neq j\}| \leq k$.

Now, suppose that $n$ samples are drawn from this underlying distribution (1) with true parameter $\Theta^*$. We further allow some samples are corrupted and not drawn from (1). Specifically, the set of sample indices $\{1, 2, \ldots, n\}$ is separated into two disjoint subsets: if $i$-th sample is in the set of "good" samples, which we name $G$, then it is a genuine sample from (1) with the parameter $\Theta^*$. On

**Algorithm 1** Trimmed Graphical Lasso in (3)
___

Initialize $\Theta^{(0)}$ (e.g. $\Theta^{(0)} = (S + \lambda I)^{-1}$)
**repeat**
    Compute $w^{(t)}$ given $\Theta^{(t-1)}$, by assigning a weight of one to the $h$ observations with lowest negative log-likelihood and a weight of zero to the remaining ones.
    $\nabla \mathcal{L}^{(t)} \leftarrow \frac{1}{h} \sum_{i=1}^{n} w_i^{(t)} X^{(i)} (X^{(i)})^\top - (\Theta^{(t-1)})^{-1}$
    *Line search.* Choose $\eta^{(t)}$ (See Nesterov (2007) for a discussion of how the stepsize may be chosen), checking that the following update maintains positive definiteness. This can be verified via Cholesky factorization (as in [23]).
    *Update.* $\Theta^{(t)} \leftarrow \mathcal{S}_{\eta^{(t)}\lambda}(\Theta^{(t-1)} - \eta^{(t)}\nabla \mathcal{L}^{(t)})$, where $\mathcal{S}$ is the soft-thresholding operator: $[\mathcal{S}_\nu(U)]_{i,j} = \text{sign}(U_{i,j})\max(|U_{i,j}| - \nu, 0)$ and is only applied to the off-diagonal elements of matrix $U$.
    Compute $(\Theta^{(t)})^{-1}$ reusing the Cholesky factor.
**until** stopping criterion is satisfied
___

the other hand, if the $i$-th sample is in the set of "bad" samples, $B$, the sample is corrupted. The identifications of $G$ and $B$ are hidden to us. However, we naturally assume that only a small number of samples are corrupted:

**(C-2)** Let $h$ be the number of good samples: $h := |G|$ and hence $|B| = n - h$. Then, we assume that larger portion of samples are genuine and uncorrupted so that $\frac{|G|-|B|}{|G|} \geq \alpha$ where $0 < \alpha \leq 1$. If we assume that 40% of samples are corrupted, then $\alpha = \frac{0.6n - 0.4n}{0.6n} = \frac{1}{3}$.

In later sections, we will derive a *robust* estimator for corrupted samples of sparse Gaussian graphical models and provide statistical guarantees of our estimator under the conditions (C-1) and (C-2).

## 2.1 Trimmed Graphical Lasso

We now propose a *Trimmed Graphical Lasso* for robust estimation of sparse GGMs:

$$\underset{\Theta \in \Omega, w}{\text{minimize}} \left\langle\!\!\left\langle \Theta, \frac{1}{h} \sum_{i=1}^{n} w_i X^{(i)}(X^{(i)})^\top \right\rangle\!\!\right\rangle - \log\det(\Theta) + \lambda\|\Theta\|_{1,\text{off}} \tag{3}$$
$$\text{s.t. } w \in [0,1]^n,\ \mathbf{1}^\top w = h,\ \|\Theta\|_1 \leq R$$

where $\lambda$ is a regularization parameter to decide the sparsity of our estimation, and $h$ is another parameter, which decides the number of samples (or sum of weights) used in the training. $h$ is ideally set as the number of uncorrupted samples in $G$, but practically we can tune the parameter $h$ by cross-validation. Here, the constraint $\|\Theta\|_1 \leq R$ is required to analyze this non-convex optimization problem as discussed in [22]. For another tuning parameter $R$, any positive real value would be sufficient for $R$ as long as $\|\Theta^*\|_1 \leq R$. Finally note that when $h$ is fixed as $n$ (and $R$ is set as infinity), the optimization problem (3) will be simply reduced to the vanilla $\ell_1$ regularized MLE for sparse GGM without concerning outliers.

The optimization problem (3) is convex in $w$ as well as in $\Theta$, however this is not the case jointly. Nevertheless, we will show later that **any** local optimum of (3) is guaranteed to be strongly consistent under some fairly mild conditions.

**Optimization** As we briefly discussed above, the problem (3) is not jointly convex but biconvex. One possible approach to solve the objective of (3) thus is to alternate between solving for $\Theta$ with fixed $w$ and solving for $w$ with fixed $\Theta$. Given $\Theta$, solving for $w$ is straightforward and boils down to assigning a weight of one to the $h$ observations with lowest negative log-likelihood and a weight of zero to the remaining ones. Given $w$, solving for $\Theta$ can be accomplished by any algorithm solving the "vanilla" graphical Lasso program, e.g. [11, 12]. Each step solves a convex problem hence the objective is guaranteed to decrease at each iteration and will converge to a local minima.

A more efficient optimization approach can be obtained by adopting a partial minimization strategy for $\Theta$: rather than solving to completion for $\Theta$ each time $w$ is updated, one performs a single step update. This approach stems from considering the following equivalent reformulation of our

objective:

$$\underset{\Theta \in \Omega}{\text{minimize}} \ \left\langle\!\!\left\langle \Theta, \frac{1}{h}\sum_{i=1}^{n} w_i(\Theta) X^{(i)}(X^{(i)})^\top \right\rangle\!\!\right\rangle - \log\det(\Theta) + \lambda\|\Theta\|_{1,\text{off}}$$

$$\text{s.\,t.\ } w_i(\Theta) = \underset{w \in [0,1]^n,\, \mathbf{1}^\top w = h}{\text{argmin}} \left\langle\!\!\left\langle \Theta, \frac{1}{h}\sum_{i=1}^{n} w_i X^{(i)}(X^{(i)})^\top \right\rangle\!\!\right\rangle, \quad \|\Theta\|_1 \le R, \tag{4}$$

On can then leverage standard first-order methods such as projected and composite gradient descent [24] that will converge to local optima. The overall procedure is depicted in Algorithm 1. Therein we assume that we pick $R$ sufficiently large, so one does not need to enforce the constraint $\|\Theta\|_1 \le R$ explicitly. If needed the constraint can be enforced by an additional projection step [22].

## 3 Statistical Guarantees of Trimmed Graphical Lasso

One of the main contributions of this paper is to provide the statistical guarantees of our Trimmed Graphical Lasso estimator for GGMs. The optimization problem (3) is non-convex, and therefore the gradient-type methods solving (3) will find estimators by local minima. Hence, our theory in this section provides the statistical error bounds on **any** local minimum measured by $\|\cdot\|_{\text{F}}$ and $\|\cdot\|_{1,\text{off}}$ norms simultaneously.

Suppose that we have some local optimum $(\widetilde{\Theta}, \widetilde{w})$ of (3) by arbitrary gradient-based method. While $\Theta^*$ is fixed unconditionally, we define $w^*$ as follows: for a sample index $i \in G$, $w_i^*$ is simply set to $\widetilde{w}_i$ so that $w_i^* - \widetilde{w}_i = 0$. Otherwise for a sample index $i \in B$, we set $w_i^* = 0$. Hence, $w^*$ is dependent on $\widetilde{w}$.

In order to derive the upper bound on the Frobenius norm error, we first need to assume the standard restricted strong convexity condition of (3) with respective to the parameter $\Theta$:

**(C-3) (Restricted strong convexity condition)** Let $\Delta$ be an arbitrary error of parameter $\Theta$. That is, $\Delta := \Theta - \Theta^*$. Then, for any possible error $\Delta$ such that $\|\Delta\|_{\text{F}} \le 1$,

$$\left\langle\!\!\left\langle \left(\Theta^*\right)^{-1} - \left(\Theta^* + \Delta\right)^{-1}, \Delta \right\rangle\!\!\right\rangle \ge \kappa_l \|\Delta\|_{\text{F}}^2 \tag{5}$$

where $\kappa_l$ is a curvature parameter.

Note that in order to guarantee the Frobenius norm-based error bounds, (C-3) is required even for the vanilla Gaussian graphical models *without* outliers, which has been well studied by several works such as the following lemma:

**Lemma 1** (Section B.4 of [22]). *For any $\Delta \in \mathbb{R}^{p \times p}$ such that $\|\Delta\|_{\text{F}} \le 1$,*

$$\left\langle\!\!\left\langle \left(\Theta^*\right)^{-1} - \left(\Theta^* + \Delta\right)^{-1}, \Delta \right\rangle\!\!\right\rangle \ge \left(\|\!|\Theta^*\|\!|_2 + 1\right)^{-2} \|\Delta\|_{\text{F}}^2,$$

thus (C-3) holds with $\kappa_l = \left(\|\!|\Theta^*\|\!|_2 + 1\right)^{-2}$.

While (C-3) is a standard condition that is also imposed for the conventional estimators under clean set of of samples, we additionally require the following condition for a successful estimation of (3) on corrupted samples:

**(C-4)** Consider arbitrary local optimum $(\widetilde{\Theta}, \widetilde{w})$. Let $\widetilde{\Delta} := \widetilde{\Theta} - \Theta^*$ and $\widetilde{\Gamma} := \widetilde{w} - w^*$. Then,

$$\left| \left\langle\!\!\left\langle \frac{1}{h}\sum_{i=1}^{n} \widetilde{\Gamma}_i X^{(i)}(X^{(i)})^\top, \widetilde{\Delta} \right\rangle\!\!\right\rangle \right| \le \tau_1(n,p)\|\widetilde{\Delta}\|_{\text{F}} + \tau_2(n,p)\|\widetilde{\Delta}\|_1$$

with some positive quantities $\tau_1(n,p)$ and $\tau_2(n,p)$ on $n$ and $p$. These will be specified below for some concrete examples.

(C-4) can be understood as a structural incoherence condition between the model parameter $\Theta$ and the weight parameter $w$. Such a condition is usually imposed when analyzing estimators with multiple parameters (for example, see [25] for a robust linear regression estimator). Since $w^*$ is defined

depending on $\widetilde{w}$, each local optimum has its own (C-4) condition. We will see in the sequel that under some reasonable cases, this condition for any local optimum holds with high probability. Also note that for the case with clean samples, the condition (C-4) is trivially satisfied since $\widetilde{\Gamma}_i = 0$ for all $i \in \{1, \ldots, n\}$ and hence the LHS becomes 0.

Armed with these conditions, we now state our main theorem on the error bounds of our estimator (3):

**Theorem 1.** *Consider corrupted Gaussian graphical models. Let $(\widetilde{\Theta}, \widetilde{w})$ be an* **any** *local optimum of $M$-estimator* (3). *Suppose that $(\widetilde{\Theta}, \widetilde{w})$ satisfies the condition* (C-4). *Suppose also that the regularization parameter $\lambda$ in* (3) *is set such that*

$$4 \max \left\{ \left\| \frac{1}{h} \sum_{i=1}^n w_i^* X^{(i)} (X^{(i)})^\top - (\Theta^*)^{-1} \right\|_\infty , \tau_2(n,p) \right\} \leq \lambda \leq \frac{\kappa_l - \tau_1(n,p)}{3R}. \qquad (6)$$

*Then, this local optimum $(\widetilde{\Theta}, \widetilde{w})$ is guaranteed to be consistent as follows:*

$$\|\widetilde{\Theta} - \Theta^*\|_{\mathrm{F}} \leq \frac{1}{\kappa_l} \left( \frac{3\lambda \sqrt{k+p}}{2} + \tau_1(n,p) \right) \quad and$$

$$\|\widetilde{\Theta} - \Theta^*\|_{1,\mathrm{off}} \leq \frac{2}{\lambda \kappa_l} \left( 3\lambda \sqrt{k+p} + \tau_1(n,p) \right)^2. \qquad (7)$$

The statement in Theorem 1 holds deterministically, and the probabilistic statement comes where we show (C-4) and (6) for a given $(\widetilde{\Theta}, \widetilde{w})$ are satisfied. Note that, defining $\mathcal{L}(\Theta, w) := \left\langle\!\!\left\langle \Theta, \frac{1}{h} \sum_{i=1}^n w_i X^{(i)} (X^{(i)})^\top \right\rangle\!\!\right\rangle - \log \det(\Theta)$, it is a standard way of choosing $\lambda$ based on $\|\nabla_\Theta \mathcal{L}(\Theta^*, w^*)\|_\infty$ (see [26], for details). Also it is important to note that the term $\sqrt{k+p}$ captures the relation between element-wise $\ell_1$ norm and the error norm $\|\cdot\|_{\mathrm{F}}$ including *diagonal entries*. Due to the space limit, the proof of Theorem 1 (and all other proofs) are provided in the Supplements [27].

Now, it is natural to ask how easily we can satisfy the conditions in Theorem 1. Intuitively it is impossible to recover true parameter by weighting approach as in (3) when the amount of corruptions exceeds that of normal observation errors.

To this end, suppose that we have some upper bound on the corruptions:

**(C-B1)** For some function $f(\cdot)$, we have $\left( \|X^B\|_2 \right)^2 \leq f(X^B) \sqrt{h \log p}$

where $X^B$ denotes the sub-design matrix in $\mathbb{R}^{|B| \times p}$ corresponding to outliers. Under this assumption, we can properly choose the regularization parameter $\lambda$ satisfying (6) as follows:

**Corollary 1.** *Consider corrupted Gaussian graphical models with conditions* (C-2) *and* (C-B1). *Suppose that we choose the regularization parameter*

$$\lambda = 4 \max \left\{ 8 (\max_i \Sigma_{ii}^*) \sqrt{\frac{10\tau \log p}{h - |B|}} + \frac{|B|}{h} \|\Sigma^*\|_\infty , f(X^B) \sqrt{\frac{\log p}{h}} \right\} \leq \frac{\kappa_l - f(X^B) \sqrt{\frac{|B| \log p}{h}}}{3R} .$$

*Then,* **any** *local optimum of* (3) *is guaranteed to satisfy* (C-4) *and have the error bounds in* (7) *with probability at least $1 - c_1 \exp(-c_1' h \lambda^2)$ for some universal positive constants $c_1$ and $c_1'$.*

If we further assume the number of corrupted samples scales with $\sqrt{n}$ at most :

**(C-B2)** $|B| \leq a \sqrt{n}$ for some constant $a \geq 0$,

then we can derive the following result as another corollary of Theorem 1:

**Corollary 2.** *Consider corrupted Gaussian graphical models. Suppose that the conditions* (C-2), (C-B1) *and* (C-B2) *hold. Also suppose that the regularization parameter $\lambda$ is set as $c\sqrt{\frac{\log p}{n}}$ where $c := 4 \max \left\{ 16 (\max_i \Sigma_{ii}^*) \sqrt{5\tau} + \frac{2a\|\Sigma^*\|_\infty}{\sqrt{\log p}} , \sqrt{2} f(X^B) \right\}$. Then, if the sample size $n$ is lower*

*bounded as*

$$n \geq \max\left\{16a^2, \left(\|\Theta^*\|_2 + 1\right)^4 \left(3Rc + f(X^B)\sqrt{2|B|}\right)^2 (\log p)\right\},$$

*then* **any** *local optimum of* (3) *is guaranteed to satisfy* (C-4) *and have the following error bound:*

$$\|\widetilde{\Theta} - \Theta^*\|_{\mathrm{F}} \leq \frac{1}{\kappa_l}\left(\frac{3c}{2}\sqrt{\frac{(k+p)\log p}{n}} + f(X^B)\sqrt{\frac{2|B|\log p}{n}}\right) \tag{8}$$

*with probability at least* $1 - c_1 \exp(-c_1' h \lambda^2)$ *for some universal positive constants* $c_1$ *and* $c_1'$.

Note that the $\|\cdot\|_{1,\mathrm{off}}$ norm based error bound also can be easily derived using the selection of $\lambda$ from (7). Corollary 2 reveals an interesting result: even when $O(\sqrt{n})$ samples out of total $n$ samples are corrupted, our estimator (3) can successfully recover the true parameter with guaranteed error in (8). The first term in this bound is $O\left(\sqrt{\frac{(k+p)\log p}{n}}\right)$ which exactly recovers the Frobenius error bound for the case without outliers (see [13, 22] for example). Due to the outliers, we have the performance degrade with the second term, which is $O\left(\sqrt{\frac{|B|\log p}{n}}\right)$. To the best of our knowledge, this is the first statistical error bounds on the parameter estimation for Gaussian graphical models with outliers. Also note that Corollary 1 only concerns on any local optimal point derived by an arbitrary optimization algorithm. For the guarantees of multiple local optima simultaneously, we may use a union bound from the corollary.

**When Outliers Follow a Gaussian Graphical Model**   Now let us provide a concrete example and show how $f(X^B)$ in (C-B1) is precisely specified in this case:

**(C-B3)** Outliers in the set $B$ are drawn from another Gaussian graphical model (1) with a parameter $(\Sigma_B)^{-1}$.

This can be understood as the Gaussian mixture model where the most of the samples are drawn from $(\Theta^*)^{-1}$ that we want to estimate, and small portion of samples are drawn from $\Sigma_B$. In this case, Corollary 2 can be further shaped as follows:

**Corollary 3.** *Suppose that the conditions* (C-2), (C-B2) *and* (C-B3) *hold. Then the statement in Corollary 2 holds with* $f(X^B) := \frac{4\sqrt{2}a\left(1+\sqrt{\log p}\right)^2 \|\Sigma_B\|_2}{\sqrt{\log p}}$.

## 4   Experiments

In this section we corroborate the performance of our Trimmed Graphical Lasso (*trim-glasso*) algorithm on simulated data. We compare against *glasso*: the vanilla Graphical Lasso [11]; the *t-Lasso* and *t\*-lasso* methods [19], and *robust-LL*: the robustified-likelihood approach of [20].

### 4.1   Simulated data

Our simulation setup is similar to [20] and is a akin to gene regulatory networks. Namely we consider four different scenarios where the outliers are generated from models with different graphical structures. Specifically, each sample is generated from the following mixture distribution:

$$y_k \sim (1-p_0)N_p(0,\Theta^{-1}) + \frac{p_0}{2}N_p(-\mu,\Theta_o^{-1}) + \frac{p_0}{2}N_p(\mu,\Theta_o^{-1}), \quad k=1,\ldots,n,$$

where $p_o = 0.1, n = 100$, and $p = 150$. Four different outlier distributions are considered:

M1: $\mu = (1,\ldots,1)^T, \Theta_o = \tilde{\Theta},$   M2: $\mu = (1.5,\ldots,1.5)^T, \Theta_o = \tilde{\Theta},$
M3: $\mu = (1,\ldots,1)^T, \Theta_o = I_p,$   M4: $\mu = (1.5,\ldots,1.5)^T, \Theta_o = I_p.$

We also consider the scenario where the outliers are not symmetric about the mean and simulate data from the following model

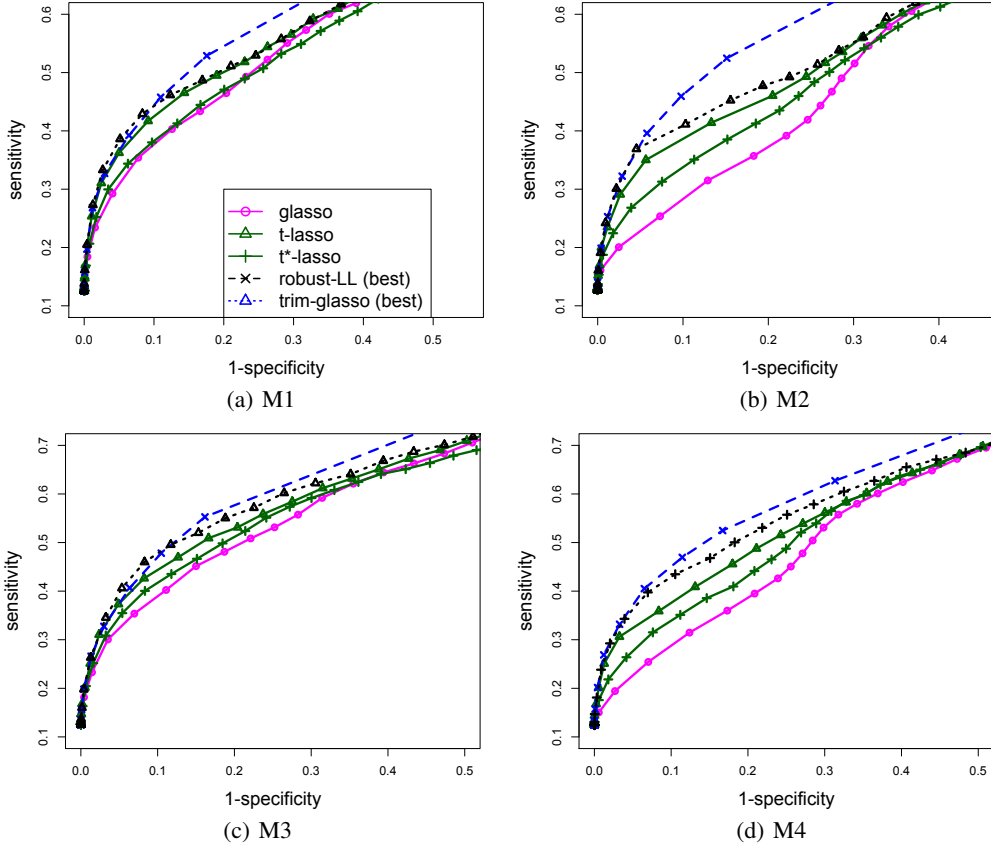

Figure 1: Average ROC curves for the comparison methods for contamination scenarios M1-M4.

$$\text{M5: } y_k \sim (1 - p_o)N_p(0, \Theta^{-1}) + p_o N_p(2, I_p), \ \ k = 1, \ldots, n.$$

For each simulation run, $\Theta$ is a randomly generated precision matrix corresponding to a network with 9 hub nodes simulated as follows. Let $A$ be the adjacency of the network. For all $i < j$ we set $A_{ij} = 1$ with probability 0.03, and zero otherwise. We set $A_{ji} = A_{ij}$. We then randomly select 9 hub nodes and set the elements of the corresponding rows and columns of $A$ to one with probability 0.4 and zero otherwise. Using $A$, the simulated nonzero coefficients of the precision matrix are sampled as follows. First we create a matrix $E$ so that $E_{i,j} = 0$ if $A_{i,j} = 0$, and $E_{i,j}$ is sampled uniformly from $[-0.75, -0.23] \cup [0.25, 0.75]$ if $A_{i,j} \neq 0$. Then we set $E = \frac{E + E^T}{2}$. Finally we set $\Theta = E + (0.1 - \Lambda_{\min}(E))I_p$, where $\Lambda_{\min}(E)$ is the smallest eigenvalue of $E$. $\tilde{\Theta}$ is a randomly generated precision matrix in the same way $\Theta$ is generated.

For the robustness parameter $\beta$ of the *robust-LL* method, we consider $\beta \in \{0.005, 0.01, 0.02, 0.03\}$ as recommended in [20]. For the *trim-glasso* method we consider $\frac{100h}{n} \in \{90, 85, 80\}$. Since all the robust comparison methods converge to a stationary point, we tested various initialization strategies for the concentration matrix, including $I_p$, $(S + \lambda I_p)^{-1}$ and the estimate from glasso. We did not observe any noticeable impact on the results.

Figure 1 presents the average ROC curves of the comparison methods over 100 simulation data sets for scenarios M1-M4 as the tuning parameter $\lambda$ varies. In the figure, for *robust-LL* and *trim-glasso* methods, we depict the best curves with respect to parameter $\beta$ and $h$ respectively. Due to space constraints, the detailed results for all the values of $\beta$ and $h$ considered, as well as the results for model M5 are provided in the Supplements [27].

From the ROC curves we can see that our proposed approach is competitive compared the alternative robust approaches *t-lasso*, *t\*-lasso* and *robust-LL*. The edge over glasso is even more pronounced for

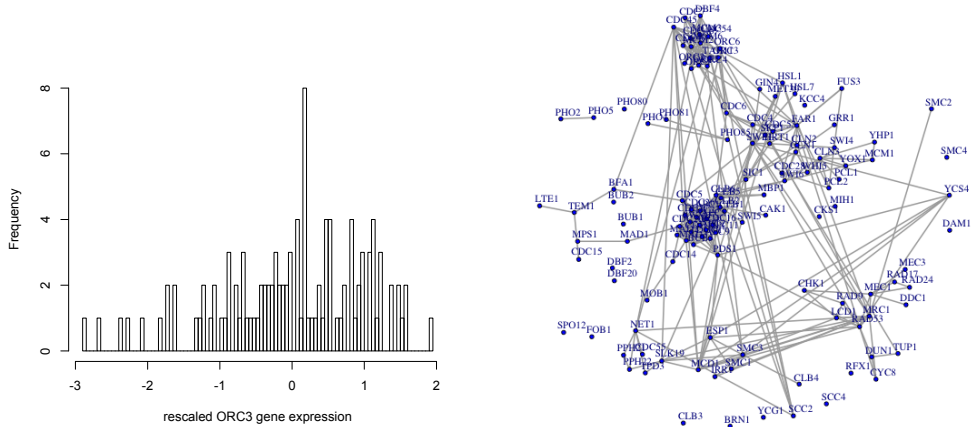

Figure 2: (a) Histogram of standardized gene expression levels for gene *ORC3*. (b) Network estimated by *trim-glasso*

scenarios M2, M4 and M5. Surprisingly, *trim-glasso* with $h/n = 80\%$ achieves superior sensitivity for nearly any specificity.

Computationally the *trim-glasso* method is also competitive compared to alternatives. The average run-time over the path of tuning parameters $\lambda$ is 45.78s for *t-lasso*, 22.14s for *t\*-lasso*, 11.06s for *robust-LL*, 1.58s for trimmed lasso, 1.04s for glasso. Experiments were run on R in a single computing node with a Intel Core i5 2.5GHz CPU and 8G memory. For *t-lasso*, *t\*-lasso* and *robust-LL* we used the R implementations provided by the methods' authors. For *glasso* we used the *glassopath* package.

## 4.2 Application to the analysis of Yeast Gene Expression Data

We analyze a yeast microarray dataset generated by [28]. The dataset concerns $n = 112$ yeast segregants (instances). We focused on $p = 126$ genes (variables) belonging to cell-cycle pathway as provided by the KEGG database [29]. For each of these genes we standardize the gene expression data to zero-mean and unit standard deviation. We observed that the expression levels of some genes are clearly not symmetric about their means and might include outliers. For example the histogram of gene *ORC3* is presented in Figure 2(a). For the *robust-LL* method we set $\beta = 0.05$ and for *trim-glasso* we use $h/n = 80\%$. We use 5-fold-CV to choose the tuning parameters for each method. After $\lambda$ is chosen for each method, we rerun the methods using the full dataset to obtain the final precision matrix estimates.

Figure 2(b) shows the cell-cycle pathway estimated by our proposed method. For comparison the cell-cycle pathway from the KEGG [29] is provided in the Supplements [27]. It is important to note that the KEGG graph corresponds to what is currently known about the pathway. It should *not* be treated as the ground truth. Certain discrepancies between KEGG and estimated graphs may also be caused by inherent limitations in the dataset used for modeling. For instance, some edges in cell-cycle pathway may not be observable from gene expression data. Additionally, the perturbation of cellular systems might not be strong enough to enable accurate inference of some of the links.

*glasso* tends to estimate more links than the robust methods. We postulate that the lack of robustness might result in inaccurate network reconstruction and the identification of spurious links. Robust methods tend to estimate networks that are more consistent with that from the KEGG ($F_1$-score of 0.23 for *glasso*, 0.37 for *t\*-lasso*, 0.39 for *robust-NLL* and 0.41 for *trim-glasso*, where the $F_1$ score is the harmonic mean between precision and recall). For instance our approach recovers several characteristics of the KEGG pathway. For instance, genes *CDC6* (a key regulator of DNA replication playing important roles in the activation and maintenance of the checkpoint mechanisms coordinating S phase and mitosis) and *PDS1* (essential gene for meiotic progression and mitotic cell cycle arrest) are identified as a hub genes, while genes *CLB3,BRN1,YCG1* are unconnected to any other genes.

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
