[Reviews · NeurIPS 2015]

Submitted by Assigned_Reviewer_1

The paper presents a method for robust inference of sparse GGM models. Theoretical guarantees on estimated models are given even when these estimates correspond to merely local optima of the relevant objective function.

The paper is clearly presented and organised logically: after introducing the problem to be solved the objective function (3) is presented. A reasonable approach to optimisation is presented - Algorithm 1. Since the objective is only bi-convex not convex no attempt is made to find a global minimum. Instead theoretical guarantees are given on *any* local minimum subject to certain conditions. Some work is done in Section 3 to provide the reader with some feeling for when these conditions will be met. Experiments with both synthetic and real data are provided.

(3) is a reasonable way of dealing with outliers/noise and it is clearly good to have the theoretical guarantees. However, I found it hard to be convinced that much had been gained - for practical problems - over existing technqiues. (The authors deserve praise for comparing against a good selection of alternatives.) The results in Fig 1 show that the proposed method is indeed "competitive" but not superior by a large margin. On the other hand doing somewhat better than alternatives *and* having some theoretical guarantees is a contribution.

Other points...

Please mention that \tau_{1}, \tau_{2} and \tau are all defined in the supplementary material - I searched hard for their definitions in the main paper!

I did not understand the sentence at line 236. What does it mean to choose a parameter "from [a] quantity"?

51 paid on -> paid to 123 if i-th sample -> if the i-th sample 125 only small number -> only a small number 242 how easy we can satisfy -> how easily we can satisfy 277 Note that -> Note that the 278 Corollary 2 recover -> Corollary 2 reveals an 285 by arbitrary optimization -> by an arbitrary optimization

286 multiple local optimum -> multiple local optima 291 from other -> from another

365 Fix j subscript. Also is -> if 463 gaussian -> Gaussian
Summary: A method with useful theoretical guarantees and competitive empirical performance is presented.

Submitted by Assigned_Reviewer_2

This paper presents a method similar to Least Trimmed Squares for robust linear regression, except for graphical lasso.

The main idea is to improve upon the original graphical lasso such that it is more robust to outliers.

The most significant contribution of this paper is the theoretical analysis showing that despite being a non-convex method, where the objective is biconvex, consistency results are given for any local minimum.

In C-4, was tao1 and tao2 ever defined?

The weakness of the this paper is in the experimental results.

It would be great if the sensitivity vs. 1-specificity figures were included in the supplemental material with the x and y axes both going from 0 to 1.

As it's currently plotted, it's difficult to assess how much better the trimmed graphical lasso is performing. From examining the plots as is, my guess is that it does not make that much difference.

Also, for the trimmed graphical lasso,

the 100h/n ratio's considered were 80,85, and 90, which means there are very few outliers.

A stronger result would be if you could show that as the outliers increase, then trimmed graphical lasso's performance begins to differentiate itself from the other methods, especially the classical graphical lasso.
Summary: This paper presents Trimmed Graphical Lasso, which is a method that induces the trimming of particular samples that are less reliable, such that the method is more robust to outliers.

The main contribution is providing statistical guarantees on the consistency of the estimator.

The experimental results show that the method is competitive with existing methods, but do not demonstrate clear superiority.

Submitted by Assigned_Reviewer_3

The paper is well-written in general.

The proposed algorithm presents a slight improvement (more significant in some cases) over the classical glasso and robust techniques, and the theoretical results give consistency guarantees.
Summary: In this paper, the authors address the problem of precision matrix estimation using sparsity-promoting methods. Based on the classical glasso, a new algorithm is proposed by adding weights to the data points. This yields a non-convex optimization problem, and the authors propose two different algorithms for solving this problem.

Of course, due to the non-convexity, the obtained solutions are local minima. However, a theoretical result is proven, showing that any local minima is guaranteed to have some consistency properties.

Author Feedback
Author rebuttal: We thank the reviewers for the constructive comments. The main goal of this paper is to present a statistical analysis of Trimmed Graphical Lasso for robust high-dimensional GGM estimation. We provide theoretical guarantees, in contrast to previously proposed approaches that do not. Experiments are included to illustrate that in addition to being theoretically sound, the simple trimming strategy is competitive with (and in certain cases superior to) more complex and time-consuming alternatives. Though beyond the scope of this paper, an in-depth empirical evaluation is an interesting and important direction for future work.

Reviewer 1
The tau's are all defined in the supplements. We will mention this in the main text.

We appreciate the insightful suggestions on the experiments. Due to space constraints we chose to fix the % of corruptions, in favor of varying the corruption strategies. As we increase the amount of corruptions the advantage of our approach indeed becomes more pronounced. We will add the corresponding ROC curves in the appendix. We believe that this holds because the trimming procedure is able to ignore completely some of the outliers in contrast to alternatives.

We will add the ROC curves with axes from 0 to 1 in the appendix (we had to zoom in in the main text due to space constraints). From the full curves we can see that as 1-specifity increases, our method remains competitive, while the other approaches all tend to merge into one curve yielding the same results as vanilla glasso. (Some of this behavior can already been observed in e.g. figure1 for M2.)

Reviewer 2
We regenerated ROC curves using precision and recall and the performance of our approach now looks even better (the curves are better "separated"). We will incorporate these in the appendix. As suggested by Reviewer 1, we also performed additional experiments with a larger % of outliers, where the advantage of our approach becomes even more significant.

Regarding yeast data: we compared with KEGG as `ground truth' and reported superior F1 score to the alternatives. We thank the reviewer for bringing the valuable yeastract to our attention. As yeastract reports documented regulations between known transcription factors and target genes, we assume that there is a relationship between 2 genes if they are regulated by the same transcription factor. Out of 313 transcription factors, 15 are found to be related to the Cell Cycle process and to regulate genes in our dataset. Treating the inferred gene adjacency matrix as ground truth, we found that our method achieves superior precision/recall curve than comparison methods. In particular, the 5-fold CV results yield F1 score of 0.21 for our approach vs 0.15 for both glasso and t*lasso, and 0.07 for robust NLL.

Reviewer 4
As suggested by Reviewer 1, we performed additional experiments with a larger % of outliers, where the advantage of our approach becomes even more significant. We will incorporate these in the appendix.

Line 236: we mean that a standard way to theoretically pick lambda is to set lambda to the l-infity norm of the gradient of the loss evaluated at \Theta*, w*. We will clarify this in the paper.

Reviewers 5,6,7
As suggested by Reviewer 1, we performed additional experiments with a larger % of outliers, where the advantage of our approach becomes even more significant. At stated above, we will incorporate these in the appendix.